# Transgenic expression of *Arabidopsis ELONGATION FACTOR-TU RECEPTOR* (AtEFR) gene in banana enhances resistance against *Xanthomonas campestris* pv. *musacearum*

Mark Adero[1,2], Jaindra Nath Tripathi [1] *, Richard Oduor[2], Cyril Zipfel [3,4], Leena Tripathi [1] *

1 International Institute of Tropical Agriculture (IITA), Nairobi, Kenya, 2 Kenyatta University, Nairobi, Kenya, 3 Department of Plant and Microbial Biology, Zurich-Basel Plant Science Center, University of Zurich, Zurich, Switzerland, 4 The Sainsbury Laboratory, University of East Anglia, Norwich Research Park, Norwich, United Kingdom

* j.tripathi@cgiar.org (JNT); l.tripathi@cgiar.org (LT)

## Abstract

Banana Xanthomonas wilt (BXW) caused by *Xanthomonas campestris* pv. *musacearum* (*Xcm*) is a severe bacterial disease affecting banana production in East and Central Africa, where banana is cultivated as a staple crop. Classical breeding of banana is challenging because the crop is clonally propagated and has limited genetic diversity. Thus, genetic engineering serves as a viable alternative for banana improvement. Studies have shown that transfer of the *elongation factor Tu receptor* gene (*AtEFR*) from *Arabidopsis thaliana* to other plant species can enhance resistance against bacterial diseases. However, *AtEFR* activity in banana and its efficacy against *Xcm* has not been demonstrated. In this study, transgenic events of banana (*Musa acuminata*) cultivar dwarf Cavendish expressing the *AtEFR* gene were generated and evaluated for resistance against *Xcm* under greenhouse conditions. The transgenic banana events were responsive to the EF-Tu-derived elf18 peptide and exhibited enhanced resistance to BXW disease compared to non-transgenic control plants. This study suggests that the functionality of *AtEFR* is retained in banana with the potential of enhancing resistance to BXW under field conditions.

## Introduction

Banana (*Musa* spp.) is a staple food for about 400 million people, a crucial food security crop, and a source of income, especially in low-income countries [1]. The Great Lakes region, including Burundi, Rwanda, Uganda, Kenya, Tanzania, and the Democratic Republic of Congo, is the largest banana producer in Africa. Bananas in this region are produced by small holder farmer mainly for local consumption. The daily consumption is estimated to be 147 kcal daily per person, which is 15 times higher than the global average and six times higher than Africa's average [2]. Unfortunately, banana production in this region is hindered by

**Funding:** This study was supported by the 2 Blades Foundation, the Gatsby Charitable Foundation, and the United States Agency for International Development (USAID). The funders had no role in study design, data collection and analysis, decision to publish, or preparation of the manuscript.

**Competing interests:** The authors have declared that no competing interests exist.

banana Xanthomonas wilt (BXW), a systemic bacterial disease caused by *Xanthomonas campestris* pv. *musacearum* (Xcm). The disease is the biggest threat to banana production in the region, with economic losses estimated to be US$ 2–8 billion over a decade [3]. Symptoms of BXW disease manifest as premature ripening and rotting of fruits, shriveling of inflorescence, wilting, and yellowing of leaves leading to the death of the infected plant. The bacterial pathogen spread by insect vectors, use of contaminated tools and infected planting materials. The disease is managed through cultural practices, including removing male buds to eliminate insect vectors, removing diseased plants, use of clean pathogen-free planting material, and disinfecting farm tools [4]. Conventional breeding of clonally propagated crops like banana is limited by the lack of genetic diversity and availability of important traits in the gene pool [5]. All cultivated banana varieties are susceptible to BXW disease; however, resistance has been observed in one of the wild banana progenitor *Musa balbisiana* belonging to the BB genome [3, 6]. Unfortunately, the B genome is laced with banana streak virus (BSV) sequences; during hybridization, recombination of integrated virus sequences sometimes results in BSV infection [7]. This has limited the use of *M. balbisiana* in conventional breeding. Notably, tolerance to *Xcm* has also been reported in *Musa acuminata* subsp. Zebrina belonging to AA genome, suggesting that tolerant traits could be present in existing banana germplasm [8]. Further, the molecular basis of disease resistance in banana progenitor *Musa balbisiana* against Xcm was investigated [6]. Comparative transcriptome analysis of BXW-resistant genotype *Musa balbisiana* and BXW-susceptible banana cultivar 'Pisang Awak' challenged with Xcm showed differentially expressed genes associated with response to biotic stress, which were mapped to the biotic stress pathways to identify genes associated with defense mechanisms. This study identified several genes involved in the activation of pathogen-associated molecular patterns (PAMP)-triggered basal defense and disease resistance (R) protein-mediated defense in *Musa balbisiana* as an early response to Xcm infection. These *Musa* defense genes can be overexpressed in the BXW-susceptible cultivars using biotechnological tools such as transgenic or genome editing for develing resistance against Xcm.

One of the approaches to enhancing plant disease resistance is by boosting their immune system [9]. Plants have evolved two main mechanisms for evading pathogens. One involves recognizing potential phytopathogens before they establish and is achieved through pattern recognition receptors (PRRs), which recognize conserved pathogen-associated molecular patterns (PAMPs), resulting in PAMP-triggered immunity (PTI). In addition, plants utilize resistance (R) proteins, that are mainly nucleotide-binding site leucine-rich repeat receptors (NLRs) that sense specific pathogen effectors leading to effector-triggered immunity (ETI) [10]. Resistance through ETI is typically race-specific and limited to plant varieties with a particular *R*-gene and specific pathogens with corresponding virulence effector. Meanwhile, PAMPs are conserved among a wide range of microbes; thus, plants tend to exhibit broad-spectrum resistance to pathogens. As PAMPs are essential for microbial survival, their evolution is slower than virulence effectors [11]. Therefore, PTI has the potential of conferring durable and broad-spectrum disease resistance compared to ETI.

Several plant PRRs have been identified and characterized. The best studied PRR is FLAGELLIN SENSING 2 (FLS2), a leucine-rich repeat receptor kinase (LRR-RK), which recognizes a N-terminal 22-amino acid motif, flg22, of bacterial flagellin [12]. Another PRR, related to FLS2, that has been extensively studied is the Brassicaceae-specific ELONGATION FACTOR TU RECEPTOR (EFR), which recognizes the N-terminal acetylated motif elf18 of bacterial elongation factor thermal unstable (EF-Tu), thereby activating plant defense response against many bacteria [13]. EF-Tu is the most copious protein in bacteria and plays an indispensable role in protein synthesis by catalyzing the binding of aminoacyl transfer RNA to the ribosome [14].

EFR plays a significant role in plant response to bacterial infection, thus, it has been widely applied to engineer crops against diverse bacterial diseases. Heterologous expression of *Arabidopsis thaliana EFR* (*AtEFR*) in *Nicotiana benthamiana* and *Solanum lycopersicum* (tomato) conferred responsiveness to elf18 peptide from diverse bacterial genera, including *Agrobacterium*, *Xanthomonas*, *Pseudomonas*, and *Ralstonia*. Furthermore, the *N. benthamiana* and tomato transgenic events exhibited enhanced resistance to different genera of bacterial pathogens [15]. So far, transgenic expression of *AtEFR* has effectively enhanced the resistance of major crops against various bacterial pathogens, including wheat (*Triticum aestivum*) against *Pseudomonas syringae* pv. *oryzae* [16], rice (*Oryza sativa*) against *Xanthomonas oryzae* pv. *oryzae* [17], potato (*S. tuberosum*) against *Ralstonia solanacearum* [18], apple (*Malus malus*) against *Erwinia amylovora* [19], and sweet orange (*Citrus sinensis*) against *Xanthomonas citri* and *Xylella fastidiosa* [20].

Previously, the transgenic banana expressing sweet pepper hypersensitive response-assisting protein (*Hrap*) or plant ferredoxin-like protein (*Pflp*) genes were developed and tested [21, 22]. These transgenic bananas showed enhanced resistance to BXW through successive crop cycles and agronomic performance comparable to non-transgenic bananas under field trials in Uganda [23]. Transgenic bananas with resistance against BXW were also developed by transforming embryogenic cell suspensions of banana cultivar 'Sukali Ndiizi' with *Xa21* gene [24]. The Xa21 transgenic banana events exhibited enhanced resistance against BXW under greenhouse conditions. Transgenic banana expressing rice *NH1* gene also showed enhanced resistance against BXW disease [25]. The bacterial pathogen can evolve resistance mechanisms to overcome the defense conferred by most single genes or single mode of action. To avoid or minimize the chances of breaking down of disease resistance, there is a need to stack several transgenes with different mode of actions in the banana event for enhanced and durable resistance to BXW disease.

In this study, transgenic banana cultivar dwarf Cavendish (*Musa* spp., AAA genomic group) constitutively expressing *AtEFR* under the control of the constitutive *CaMV35S* promoter were developed and evaluated against *Xcm* under controlled greenhouse conditions. The transgenic banana events showed enhanced resistance to BXW disease, as exhibited by significantly reduced wilting incidences and a lower bacterial population compared to the non-transgenic control plants. Assessment of defense-related gene expression and oxidative burst assay further revealed that the transgenic events gained responsiveness to elf18, indicating that *AtEFR* activity is retained after its transfer and integration into the banana genome.

## Materials and methods

### Plant material

Embryogenic cell suspensions (ECS) of banana cultivar dwarf Cavendish were used for transformation. The ECS were initiated from meristematic shoot tissues using the protocol described by Tripathi et al. [26] and maintained at 28±2°C on a rotary shaker at 95 rpm in the dark.

### Generation of transgenic plants

The plasmid construct pBIN19g-35S::AtEFR [15], harboring *ELONGATION FACTOR TU RECEPTOR* gene from *Arabidopsis thaliana* (*AtEFR*) and *neomycin phosphotransferase II* (*nptII*) gene for kanamycin selection was transformed into *Agrobacterium tumefaciens* strain EHA105 by electroporation. The plasmid construct was validated through restriction enzyme digestion and PCR analysis using *AtEFR*-specific primers before using it for transformation. Banana ECSs were transformed as per the method described previously [26]. *Agrobacterium*

infected ECSs were regenerated on a selective medium supplemented with 100 mg/l kanamycin following the protocol described by Tripathi et al. [26]. All regenerated putative transgenic events were maintained and multiplied on proliferation medium at 28±2°C for a 16-/ 8-h light/dark photoperiod under fluorescent tube lights. The putative transgenic events were characterized for the presence of the transgene. The transgenic shoots were transferred to the rooting medium. The well-rooted events were transferred to soil in the greenhouse for disease evaluation.

## Molecular analysis of transgenic events

**Polymerase chain reaction analysis.** Genomic DNA was extracted from the putatively transformed events using a cetyltrimethylammonium bromide (CTAB) method [27]. PCR was performed in a 25-μL reaction volume containing 2.5 μL 10X PCR buffer, 0.3 μL dNTPs, 1μL of 10 μM reverse and forward *AtEFR* primers, 2 μL genomic DNA (100ng/μL), 0.2 μL Taq DNA polymerase (Qiagen, Germany), and 18 μL nuclease-free water. Thermocycler conditions were as follows: initial denaturation at 95°C for 5 min, followed by 35 cycles of denaturation at 94°C for 30 s, annealing at 60°C for 30 s, and extension at 72°C for 45 s, then final extension at 72°C for 7 min. Genomic DNA from non-transgenic control plant and pBIN19g-35S::*AtEFR* plasmid were used as negative and positive controls, respectively. The primers [forward 5′CGGGAATCTTGTAAGCCTGC 3′ and reverse 5′GCACCCTTCCCTCAAACTTG 3′] amplifying 635 base pairs (bp) region within the *AtEFR* gene were used for PCR analysis. The amplified products were run on a 1% agarose gel (Duchefa, Netherlands) stained with GelRed® (Biotium, San Francisco, USA) and visualized under ultraviolet light.

**Southern blot analysis.** Southern blot analysis was performed as per the method described by Tripathi et al. [23]. Briefly, 10 μg of genomic DNA from each sample was restricted with *BamH1* for 12 h. The DNA samples, including plasmid pBIN19g-35S::*AtEFR* and genomic DNA sample from a control non-transgenic plant, were run for a 0.8% agarose gel at 50 V. The gel stained with GelRed® was viewed under ultraviolet light to confirm the digestion. The restricted DNA was denatured, then blotted onto a positively charged membrane (Roche Diagnostics, West Sussex, UK) and fixed using ultraviolet cross-linking. The blots were then hybridized with a digoxigenin (DIG) PCR-labeled 635-bp *AtEFR*-specific probe. Hybridization and probe detection was performed using a DIG Luminescent Detection Kit for Nucleic Acids (Roche Diagnostics, UK) as per the manufacturer's protocol.

## Plant growth analysis

Eight PCR positive transgenic events and control plants were randomly selected for the plant growth analysis. Three replicates of well-rooted plants for each event were transferred to soil in small plastic cups and acclimatized for 30 days in a humidity chamber, then transferred to bigger pots and grown in the greenhouse for 90 days at 25–30°C. The growth parameter data, including plant height, pseudostem girth, number of functional leaves, and length and width of the middle leaf, were recorded from 90-day-old plants. The total leaf area was calculated using the formula below [28].

Total leaf area = 0.8 ⊆ L⊆ W ⊆N

In which, L = Length of the middle leaf, W = width of the middle leaf, and N = total number of leaves in the plant.

## Greenhouse evaluation of transgenic events for resistance to BXW disease

Three replicates of each transgenic event and non-transgenic control were evaluated for resistance against *Xcm* under greenhouse conditions. A total of 31 transgenic events were used for

the disease assay. The transgenic events and non-transgenic control plants were arranged in a completely randomized design. The culture of *Xcm* (Ugandan isolate, sublineage 2) that met the four Koch postulates were cultured in YPGA medium (0.5% yeast extract, 0.5% peptone, 1% glucose, and 0.8% micro agar) for 48 h at 28˚C. A single colony was aseptically isolated and further cultured for 48 h in 50 mL of YPG medium at 28˚C in an incubator shaker (200 rpm). Subsequently, the liquid culture was centrifuged at 4000 rpm for 15 min and the pellet resuspended in sterile distilled water. The suspension concentration was adjusted to $OD_{600nm}$ of 1 using sterile distilled water. The second open functional leaf of 90-day-old potted plant was inoculated with 100 μL of the bacterial suspension using an insulin syringe. The plants were maintained in the greenhouse under observation, and symptoms were recorded as they occurred for 60 days post-inoculation (dpi). The data collected included number of days for appearance of the first symptom, number of days for complete wilting, disease severity, and the number of leaves showing symptoms at 60 dpi. The data was used to calculate percent resistance compared to control non-transgenic plants using the formula below [27]:

Resistance (%) = (Reduction in wilting of transgenic event/number of leaves wilted in the control plant) ⊆100

In which, reduction in wilting was the total number of leaves minus the number of leaves wilted.

Disease severity was rated on a scale of 0–5 (0- no signs of the disease, 1- a single leaf showing symptom, 2-two to three leaves with symptoms, 3- four to five leaves with symptoms, 4- all the leaves have symptoms but plant not dead, 5- complete death).

Plants were categorized as resistant if they did not show any symptom, partially resistant if the symptoms did not spread to all the leaves, and susceptible if the symptoms spread to all the leaves, causing complete wilting or death of the plant.

## In planta bacterial population analysis

For the bacterial population study, two transgenic events (T5 and T7) that exhibited enhanced resistance against *Xcm* under greenhouse conditions, along with control non-transgenic plants, were used for this experiment. Three replicates of 90-day-old potted plants were inoculated with *Xcm* culture as described previously [21]. Leaf-mid rib sections (1 cm) of the inoculated leaves were collected at 0, 3, 6, 9, 12, and 15 pdi. The samples were ground in 15-mL falcon tubes after adding 2 mL of YPG medium and incubated in an incubator shaker (200 rpm) for 1 h at 28˚C. Five serial dilutions of the sample suspensions were spread on YPGA medium supplemented with 50 mg/L cephalexin to select for *Xcm* and cultured at 28˚C for 48 h. Samples from each line were cultured in triplicate for every time point. The bacterial population was determined by counting colonies for each dilution and described using growth curve analysis.

## Transgene expression analysis

To determine the relative gene expression of various transgenic events compared to control non-transgenic plant, total RNA was extracted from the leaves of two-week-old shoots using RNeasy plant mini kit (Qiagen, Germany) according to the manufacturer's instructions. The RNA quality and concentration were determined using NanoDrop 2000 (Thermo Fisher Scientific). And 1 μg of the total RNA was reverse transcribed into cDNA using Luna script RT Supermix (New England Biolabs) as per the manufacturer's protocol. The cDNA templates were then used for PCR amplification of the *AtEFR* gene transcript. The PCR was performed in a 25-μL reaction volume containing 2.5 μL10X PCR buffer, 0.3 μL dNTPs, 1μL of 10 μM reverse and forward *AtEFR* primers (S1 Table), 2 μL cDNA, 0.2 μL Taq DNA polymerase, and

18 μL nuclease-free water under the following conditions: initial denaturation at 95°C for 5 min followed by 35 cycles of denaturation at 94°C for 30 sec, annealing at 55°C for 30 sec, and extension at 72°C for 1 min, then final extension at 72°C for 7 min. *Musa 25s* ribosomal transcript was used as an internal control to check the quality of the cDNA. qRT-PCR was performed using Luna® Universal qPCR Master Mix (New England Biolabs) on a Quanta Studio Real-Time PCR System (Applied Biosystems, Foster City, CA) under the following conditions: initial denaturation at 95°C for 5 min followed by 40 cycles of denaturation at 94°C for 30 s, annealing at 60°C for 30 s, and extension at 72°C for 1 min. The reaction mixture was in a 20 μl reaction volume consisting of 10 μl Luna® Universal qPCR Master Mix, 0.2 μl forward and reverse primers (S1 Table), 5 μl cDNA, and 4.6 μl nuclease-free water. Each sample consisted of three biological replicates. *Musa 25s* ribosomal transcript served as an internal control for the normalization of gene expression. The relative levels of the *AtEFR* gene were analysed using Livak & Schmittgen method [29].

## Expression analysis of defense-related genes

The leaves of two-week-old shoots were infiltrated with 250 nM elf18 peptide solution for 60, 120, and 180 min. Total RNA was extracted and qRT-PCR was performed as described in the above section to check the relative expression of defense-related genes (*MaWRKY-22 like*, *MaPR1-like*, *MaPR2-like*, *MaPR3-like*, *MaPR4-like*, and *MaPR5-like*) in the transgenic events compared to control non-transgenic events. qRT-PCR was performed with initial denaturation at 95°C for 10 min followed by 40 cycles of denaturation at 95°C for 15 s, annealing at 60°C for 30 s, and extension at 72°C for 1 min. Each sample consisted of three biological replicates. *Musa 25s* ribosomal transcript was used as an internal control for the normalization of gene expression. The primers used are presented in S1 Table.

## ROS production assay

Hydrogen peroxide production in the transgenic events after elf18 infiltration was determined through histochemical staining assay using DAB (3,3'-diaminobenzidine) solution [30]. Briefly, freshly opened leaves of 14-day-old *in vitro* plantlets were infiltrated with 100 μL of 250 nM elf18 peptide dissolved in sterile distilled water. The leaves were left to incubate for 2 h, after which they were harvested and submerged in DAB solution in 15-mL falcon tubes wrapped with aluminum foil. The samples were incubated in DAB solution for 5 h on a shaker (70 rpm) at room temperature. Following incubation, DAB solution was removed, and samples were bleached with a solution (ethanol: acetic acid: glycerol = 3:1:1) for 15 min in the water bath at 95°C to remove chlorophyll. The bleaching solution was refreshed thrice, and the samples were further incubated for 30 min at room temperature. The samples were then removed from the bleaching solution, dried on paper towels, and photographed using SMZ1500 stereomicroscope (Carlsbad, CA, USA) attached with high zoom Nikon camera. The photomicrographs were analyzed using image J software (National Institutes of Health, USA) to compare the browning intensities.

## Statistical analysis

Data analysis was performed using Minitab Statistical Software, version 17 (Pennsylvania, USA). Differences in disease resistance and plant growth characteristics between various transgenic events and non-transgenic control plants were analyzed using one-way analysis of variance (ANOVA) and means separated by *Fisher's* HSD test. Statistical significance was determined at p≤0.05.

## Results

### Generation of transgenic events

Embryogenic cell suspensions (ECSs) of banana cultivar dwarf Cavendish were co-cultivated with *Agrobacterium tumefaciens* strain EHA105 harboring the binary vector pBIN19g:35S:: *AtEFR* [15] for three days. *Agrobacterium*-infected ECSs turned from cream to brown color during co-cultivation. Browning intensified when the cells were transferred to embryo development medium (EDM) supplemented with 100 mg/L kanamycin for selecting transformed embryogenic cells. After about 30 days of culture on the same medium, white embryos began to appear on the surface of the dark heap of cells (Fig 1A). The white embryos increased in number and size when the cells were transferred to fresh EDM. After 14 days of culture on

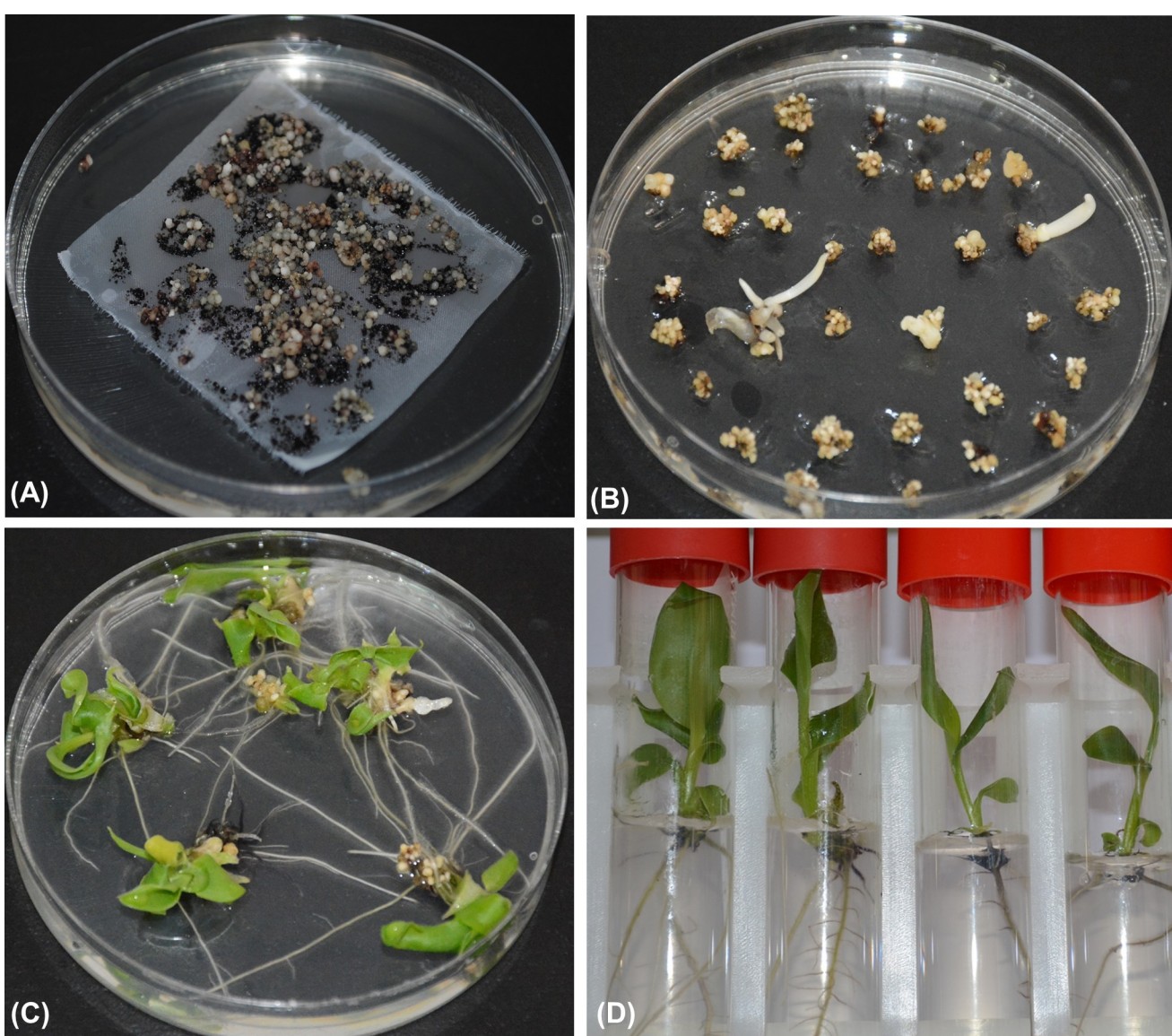

**Fig 1. Generation of transgenic banana events expressing *AtEFR* gene.** (A) Embryogenic cells in selective embryo development medium (EDM), (B) Embryos maturing and germinating on selective embryo maturation medium (EMM), (C) Germination of embryos in selective embryo germination medium (EGM), (D) Complete plantlets on selective proliferation medium (PM).

selective embryo maturation medium (EMM), a few embryos germinated into plantlets, but most continued to expand on the same medium without developing any organized structures (Fig 1B). Most embryos germinated into complete plantlets on a selective embryo germination medium (EGM) after 20–30 days of culture (Fig 1C). After germination, the shoots elongated when transferred to proliferation medium (PM) supplemented with 100 mg/L kanamycin (Fig 1D). In total, 32 putative transgenic events were regenerated from three separate experiments.

## Molecular characterization of transgenic events

The regenerated putative transgenic events were validated for the presence and integration of the transgene by PCR and Southern blot analysis, respectively. The PCR analysis using *AtEFR*-specific primers revealed an amplicon of the expected size (635 bp) (Fig 2A), confirming the presence of the *AtEFR* gene in the transgenic events. No amplification was observed in control non-transgenic plants. Further, 14 PCR-positive events were assessed using Southern blot analysis to confirm transgene integration and copy number. Southern hybridization of *BamH1*-digested genomic DNA using an *AtEFR*-specific probe confirmed the integration of the transgene in the plant genome with different hybridization profiles, indicating random insertion of the transgene in the genome of the tested events. The copy number of the transgene incorporated in the different events ranged from at least one to multiple (Fig 2B). No transgene integration was detected in the non-transgenic control plants.

## Plant growth analysis

To assess growth parameters of the generated transgenic events, plant height, pseudostem girth, total leaf area and number of functional leaves were evaluated in eight randomly selected

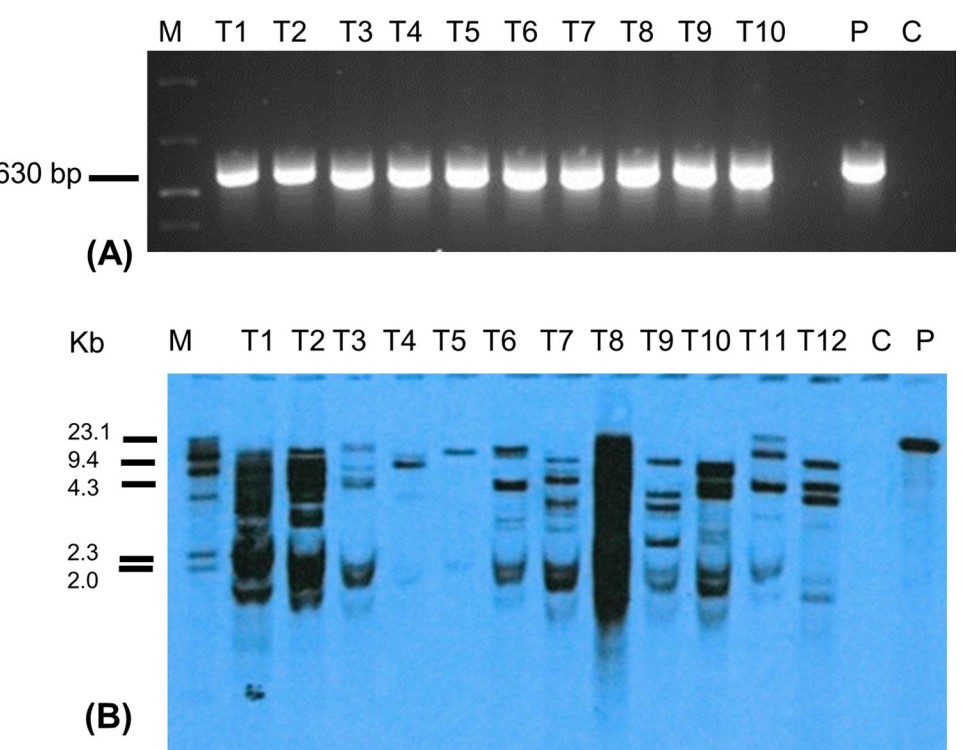

**Fig 2. Molecular analysis of putatively transgenic banana events expressing *AtEFR* gene.** A) Polymerase chain reaction assay to confirm the presence of *AtEFR* gene, B) Southern blot analysis to confirm the integration of *AtEFR* gene. M; molecular marker, C; Control, P; plasmid.

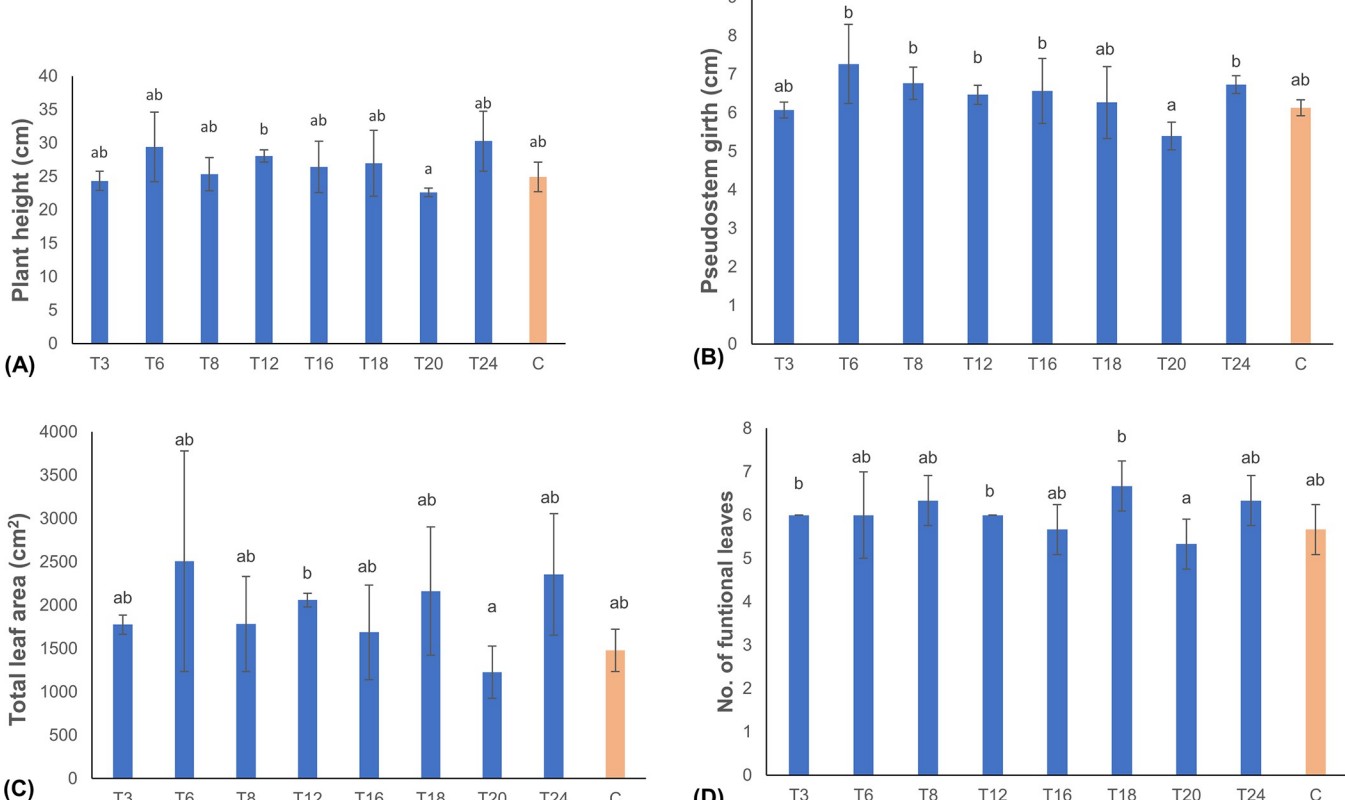

**Fig 3. Growth analysis of transgenic banana events expressing *AtEFR* gene compared to non-transgenic control plants.** (A) Plant height, B) Pseudostem girth, (C) Number of functional leaves, (D) Total leaf area. Data are presented as means ± standard deviation. Bars with different letters are significantly different at p<0.05 according to Fisher's HSD test (n = 3).

transgenic events and non-transgenic control plants. All plants showed normal growth with no morphological abnormalities. No significant differences (p<0.05) in plant height, pseudostem girth, and total leaf area were observed between the transgenic events and the non-transgenic control plants (Fig 3A–3D), indicating that overexpression of the *AtEFR* gene in transgenic banana did not alter plant growth.

## Evaluation of transgenic banana events for resistance to banana Xanthomonas wilt

Thirty-one transgenic events were evaluated through a greenhouse bioassay for disease resistance compared to non-transgenic control plants. Significant differences were observed among the transgenic and non-transgenic control events regarding the number of days to the appearance of the first symptom and the number of days to complete wilting of plants. BXW disease symptoms, such as necrosis and wilting of leaves, appeared in non-transgenic control plants at 16 dpi, whereas transgene-positive plants showed symptoms at only 18 to 42 dpi (Table 1). None of the transgene-positive plants showed complete resistance to BXW disease. Even though several transgene-positive plants showed disease symptoms, disease progression was significantly slower than in control plants. Complete wilting was observed in control plants after an average of 22 dpi, compared to transgene-positive plants, which started to collapse 25 dpi, with the majority dying 30 dpi. Out of the 31 events evaluated, 18 events were found to exhibit partial resistance (50–75%) compared to control non-transgenic plants (Table 1, Fig 4A & 4B).

**Table 1. Greenhouse evaluation of transgenic events of banana cultivar 'dwarf Cavendish' expressing *AtEFR* gene for resistance against *Xanthomonas campestris* pv. *musacearum*.**

| Line number | Mean number of days for appearance of first symptoms | Mean number of days for complete wilting | Percent resistance | Disease rating |
|---|---|---|---|---|
| T1 | 22.5±7.8[bc] | NCW | 75.4±5.4[bc] | PR |
| T2 | 27.0±2.83[b] | NCW | 66.7±9.5[b] | PR |
| T3 | 29.5±0.7[b] | 42.5±6.4[a] | 33.3±10.5[a] | S |
| T4 | 36.0±2.83[a] | NCW | 66.7±10.5[b] | PR |
| T5 | 20.5±2.12[bc] | NCW | 75.0±6.9[bc] | PR |
| T6 | 42.5±2.12[a] | NCW | 58.3±7.8[ab] | PR |
| T7 | 20±1.5[bc] | NCW | 66.7±9.5[b] | PR |
| T8 | 18±2.1[c] | NCW | 50 ±10.5[ab] | PR |
| T9 | 21.0±2.83[bc] | 33.5±2.12[ab] | 33.3±10.5[a] | S |
| T10 | 22.0±0.7[bc] | NCW | 71.4±9.0[bc] | PR |
| T11 | 27.0±0.0[b] | NCW | 66.7±10.5[b] | PR |
| T12 | 28.5±14.8[b] | 31.0±2.6[abc] | 38.1±9.5[a] | S |
| T13 | 18.3±1.5[c] | 26.3±2.1[bc] | 0[a] | S |
| T14 | 21.0±2.8[bc] | NCW | 66.7±10.5[b] | PR |
| T15 | 24.0±1.4[bc] | 38.0±2.6[ab] | 47.6±9.9[ab] | S |
| T16 | 21.5±2.1[bc] | 29.0±2.8[bc] | 33.3±10.5[a] | S |
| T17 | 20±3.1[bc] | NCW | 50±9.9[ab] | PR |
| T18 | 21.0±2.8[bc] | NCW | 66.7±10.5[b] | PR |
| T19 | 22.0±2.1[bc] | NCW | 66.7±10.5[b] | PR |
| T20 | 20.5±3.5[bc] | 33.0±2.8[ab] | 33.3±9.5[a] | S |
| T21 | 28.5±10.6[b] | 43.0±14.1[a] | 33.3±9.9[a] | S |
| T22 | 19.0±0.0[bc] | 25.0±2.8b[c] | 33.3±7.5[a] | S |
| T23 | 29.0±9.9[b] | NCW | 66.7±9.5[b] | PR |
| T24 | 24.0±4.2[bc] | NCW | 66.7±6.5[b] | PR |
| T25 | 23.0±0.0[bc] | 34±2.6[ab] | 41.7±9.0[ab] | S |
| T26 | 28.3±2.1[b] | NCW | 66.7±10.5[b] | PR |
| T27 | 25.5±3.1[b] | NCW | 66.7±10.5[b] | PR |
| T28 | 21.0±1.8[bc] | NCW | 66.7±10.5[b] | PR |
| T29 | 25.5±2.8[b] | 35±2.0[ab] | 33.3±9.5[a] | S |
| T30 | 27.0±0.0[b] | 32±2.8[ab] | 33.3±9.5[a] | S |
| T31 | 22.5±0.7[bc] | NCW | 66.7±10.5[b] | PR |
| Control | 16.3±1.2[c] | 22.0±2.0[c] | 0[a] | S |

Means followed by the same superscript letter in the same column are not significantly different according to *Fisher HSD* test at $p<0.05$

Data are presented as means±standard diviation (SD). NS; no symptoms

NCW; no complete wilting

S; suseptible

PR; partial resistance.

## Testing transgenic banana events for oxidative burst

To assess whether the transgenic events exhibited enhanced production of reactive oxygen species upon pathogen infection, the leaves of transgenic event T5 and non-transgenic control were infiltrated with either sterile water or 250 nM elf18 for 2 hpi, followed by diaminobenzidine (DAB) staining to detect the accumulation of hydrogen peroxide. Positive-transgenic plants infiltrated with elf18 exhibited more intense browning than the mock-treated control and transgenic plants (Fig 5A & 5B) suggesting that the expression of AtEFR confers elf18 perception and subsequent activation of early signaling immune outputs in the transgenic plants.

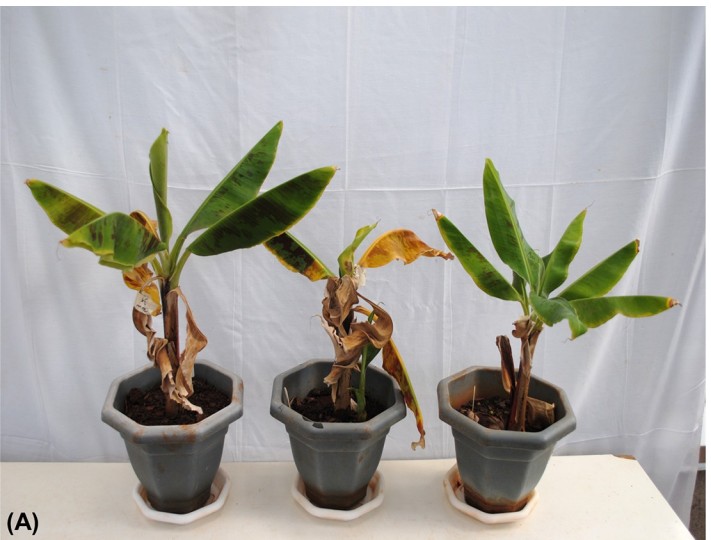
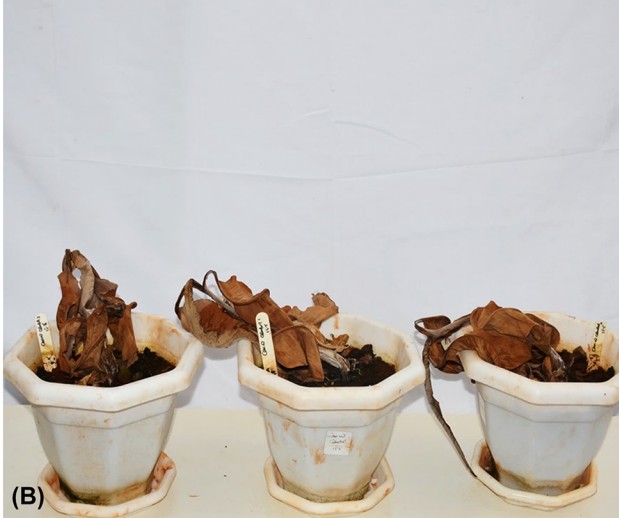

**Fig 4. Evaluation of transgenic banana events expressing *AtEFR* gene and non-transgenic control for resistance to BXW disease.** (A) Transgenic banana event T7 showing partial resistance to BXW disease, (B) Non-transgenic control plants showing complete wilting due to BXW disease. Photographs were taken at 60 days post inoculations (dpi).

## Relative expression of *AtEFR* gene among transgenic events

The relative expression of the *AtEFR* transgene was assessed through reverse-transcriptase (RT) quantitative PCR (RT-qPCR) assays in seven transgenic events compared to non-transgenic control plants. The seven transgenic events were selected based on their level of disease resistance in the glasshouse, ranging from 33% resistance in T3 event to 75% resistance in T1 and T5 event (Table 1). Variations in transcript levels were observed between the various transgenic events tested. For example, T5 event showed the highest level of *AtEFR* gene

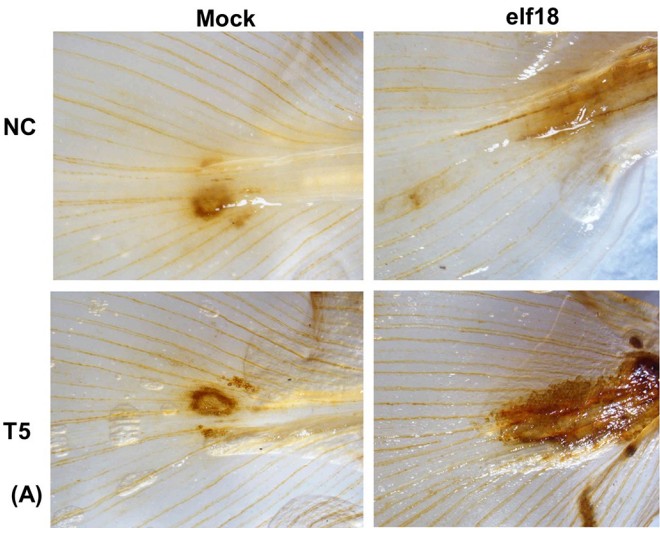
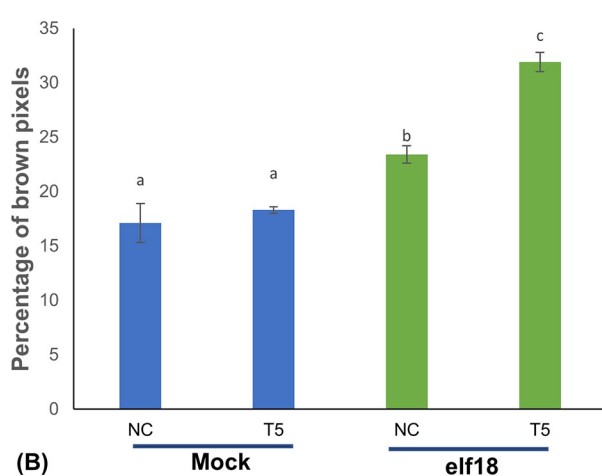

**Fig 5. ROS production assay in transgenic event T5 and non-transgenic control (NC) after elf18 infiltration followed by diaminobenzidine (DAB) staining.** A) Photomicrographs showing the difference in browning intensity between the leaves of transgenic events compared to non-transgenic control plants, (B) Graphical representation of browning intensity of non-transgenic control and transgenic leaves after elf18 infiltration followed by DAB staining. Error bars represent the standard error of the mean of three independent biological replicates.

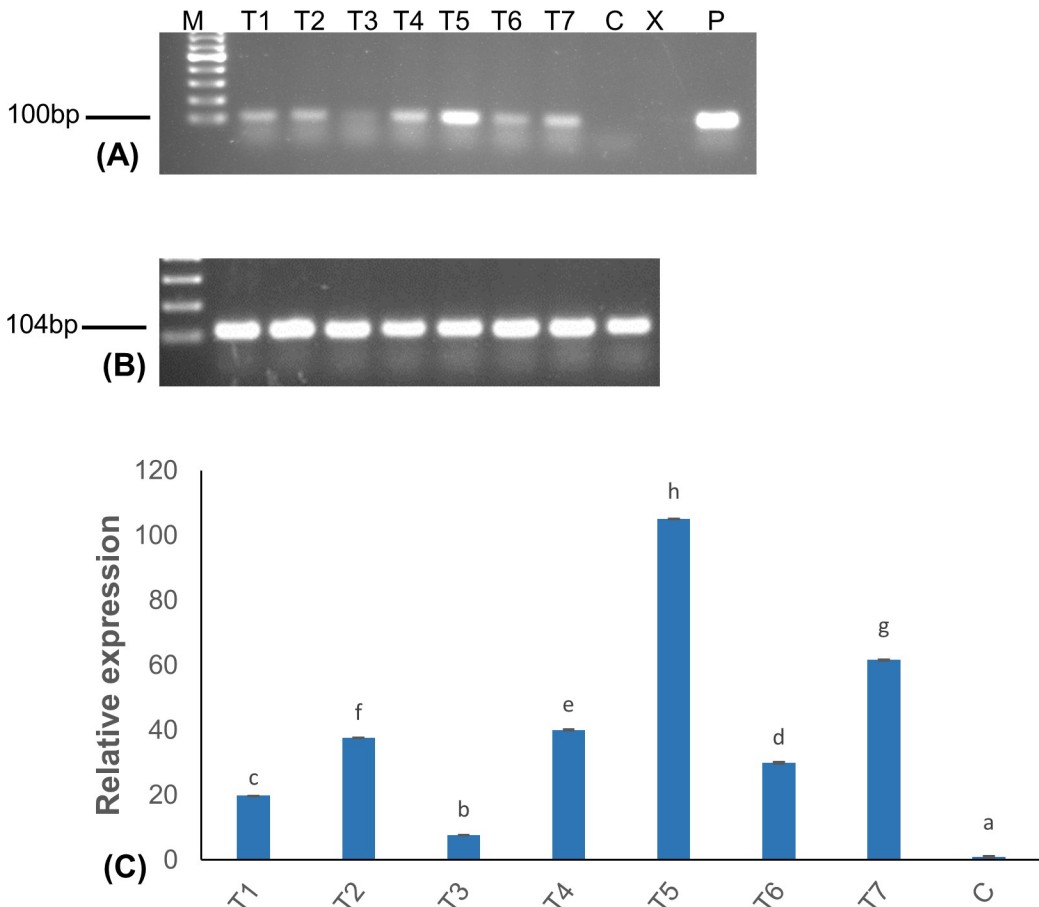

**Fig 6. Relative expression of *AtEFR* gene in transgenic banana events.** (A) Reverse-transcriptase (RT)-PCR products corresponding to 100 bp *AtEFR* gene fragment, (B) RT-PCR products corresponding to *Musa 25s* gene (internal control), (C) Transcription level of *AtEFR* in the transgenic bananas as determined using qRT-PCR. Data shown are fold induction relative to the non-transgenic control and were normalized against the internal control gene *Musa 25s*. Data presented are mean± standard deviation (SD) of three replicate experiments. Bars with different letters are significantly different at p<0.05 according to *Fisher's* HSD test. M; 100-base pair marker; T; Transgenic event expressing *AtEFR*, C; non-transgenic control.

expression, whereas the lowest gene expression was observed in T3 event (Fig 6A–6C). No *EFR* expression was detected in non-transgenic control plants.

## Relative expression of defense-related genes in transgenic banana events

The transgenic T5 event with 75% resistance was infiltrated with 250 nM elf18, and the relative expression of six selected defense marker genes (*MaWRKY-22 like*, *MaPR1-like*, *MaPR2-like*, *MaPR3-like*, *MaPR4-like*, and *MaPR5-like*) was evaluated at 1, 2, and 3 h time points, relative to the water-treated replicates. Among the tested defense-related genes, *MaWRKY-22-like*, *MaPR1-like* and *MaPR3-like* were significantly upregulated in *AtEFR* transgene-positive plants relative to non-transgenic control plants. *MaWRKY-22-like* was highly expressed mainly at 1h post-infiltration (hpi), after which its expression reduced significantly. At 3 hpi, its expression was not substantially different from that of the non-transgenic control. Similarly, *MaPR3-like* was upregulated at all the time points, but significantly at 1 hpi. Meanwhile, the expression of *MaPR1-like* and *MaPR2-like* was significantly higher at 3 hpi. Notably, no significant increase

in the expression of *MaPR4-like* and *MaPR5-like* was observed at all time points, compared with the non-transgenic control (Fig 7A–7C).

## Bacterial population analysis of transgenic banana events

In planta bacterial population analysis was performed with two transgene-positive plants (T5 and T7) and non-transgenic control plants to determine differences in the *Xcm* growth rate post inoculation. Samples were collected from each plant at specific time points following inoculation with *Xcm* and cultured on a YPGA medium to examine the bacterial population. No significant difference in bacterial titers were observed between the control and transgenic plants up to 3 dpi (Fig 8A). However, a reduction in bacterial titers was observed in transgene-positive plants compared to non-transgenic control at 6, 9, 12, and 15 dpi (Fig 8A). Also, significant differences in the bacterial population were observed between the two transgenic events assayed. Transgenic event T7 exhibited a significantly lower level of bacterial population compared to event T5 at 15 dpi (Fig 8B).

## Discussion

BXW disease continues to be the primary threat to banana production in the Great Lakes region of East Africa since its first report in Ethiopia. Considering that most bananas grown in

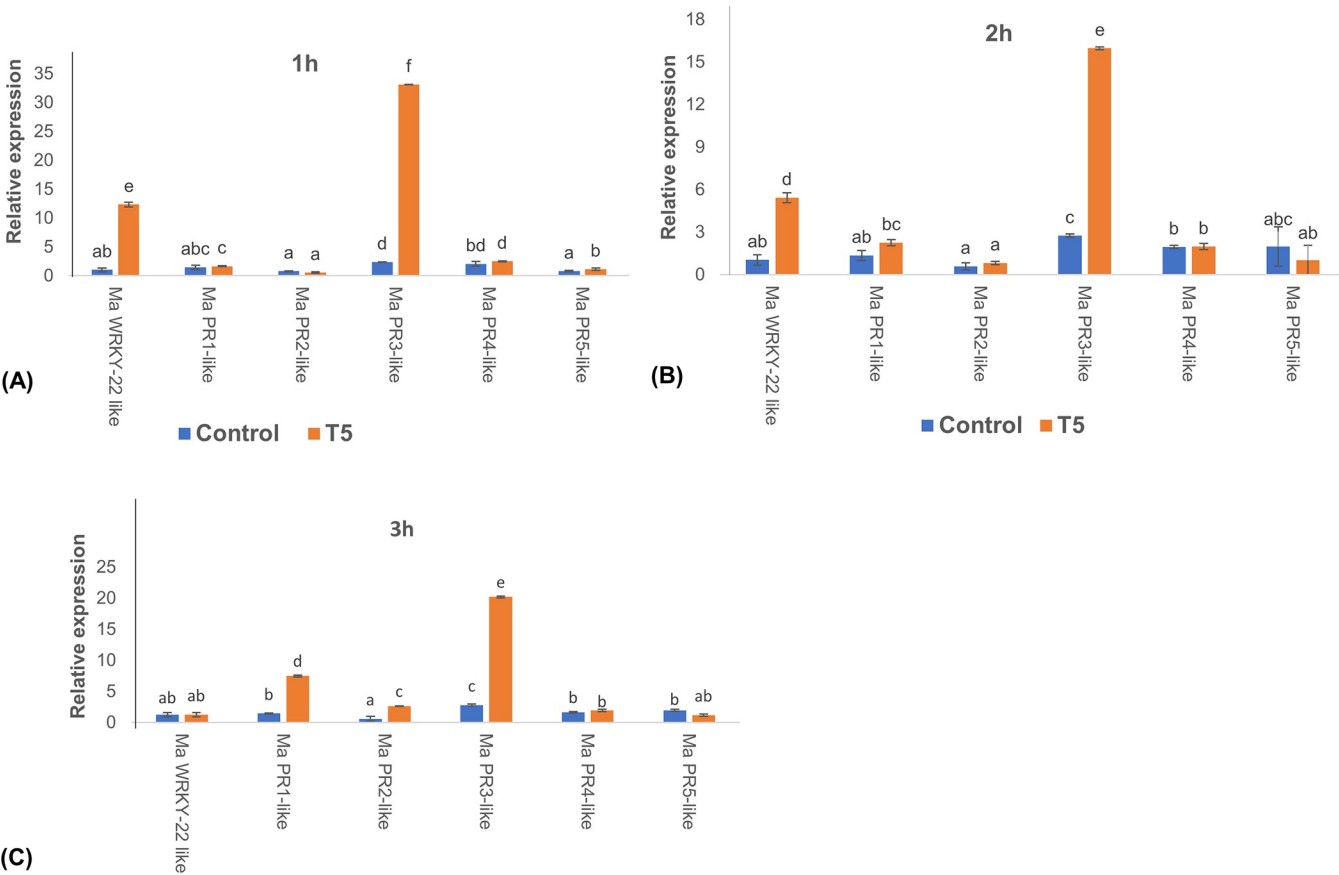

**Fig 7. Relative expression of defense related genes in transgenic bananas expressing *AtEFR* upon activation with elf18 peptide.** Expression profile of defense-related genes induced at (A) 1 h post inoculation (hpi), (B) 2 hpi, and (C) 3 hpi following infiltration with 250 nM elf18 peptide in transgenic line T5 and non-transgenic control as revealed by qRT-PCR. Results are presented as fold induction relative to water treatment and normalized to internal control *Musa 25s* gene. Data presented are mean values ± standard deviation (SD) from three replicate experiments. Bars with different letters are significantly different at p<0.05 according to *Fisher's* HSD test.

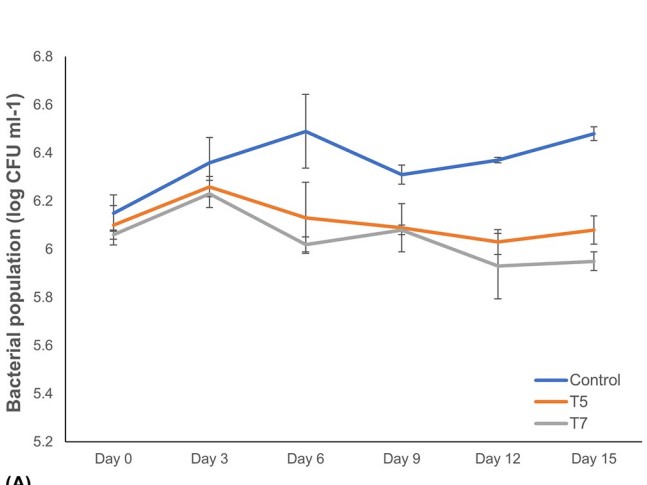
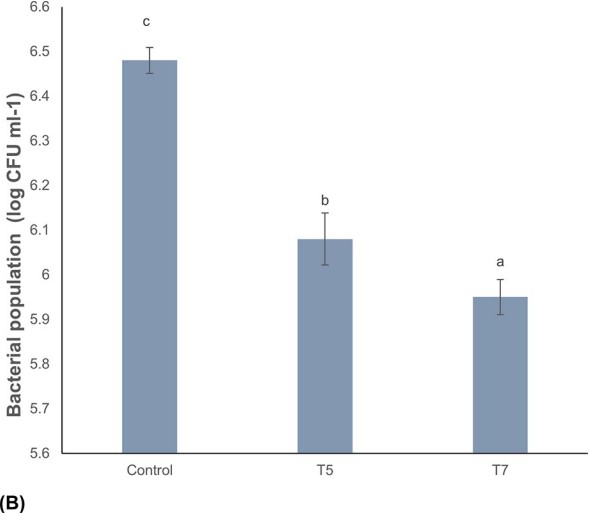

**Fig 8. Bacterial population analysis in two transgenic events (T5 and T7) and non-transgenic control plants post inoculation with *Xanthomonas campestris pv. musacearum* (*Xcm*).** (A) Bacterial population analysis at 0 to 15 dpi with *Xcm*, (B) Bacterial population at 15 dpi. Data are presented as means ± standard deviations. Bars with different letters are significantly different at p<0.05 according to *Fisher's* HSD test (n = 3).

these regions are mainly for subsistence, the impact can be severe. Developing banana varieties with disease resistance through conventional breeding is possible; however, the process has many limitations [31]. For example, most breeding programs use fertile diploids to hybridize triploid varieties or tetraploid intermediates. However, given that most cultivated varieties are triploids with low female and male fertility levels, it is not viable to repeatedly backcross the primary hybrid to the original variety. As such, the introgression of new traits into elite varieties is not practical. Also, being a clonally propagated crop, there are limited desirable traits in banana gene pool. In this case, there are no fertile diploids with resistance to BXW, except *Musa balbisiana*, which is not preferred due to its association with banana streak virus susceptibility [5].

The exploitation of natural plant defenses for crop improvement through genetic engineering is increasingly becoming an attractive option for improving clonally propagated crops like banana. Transferring PRRs across plant species that would not naturally interbreed is relatively a recent crop improvement approach for developing disease resistance [32]. Herein, the *EFR* gene from *A. thaliana* (*AtEFR*) was overexpressed in the susceptible banana cultivar 'dwarf Cavendish,' and the transgenic events were evaluated for their response against *Xcm* in greenhouse conditions. These assays revealed that transgenic bananas expressing *AtEFR* exhibited enhanced resistance against *Xcm* compared to non-transgenic controls.

Heterologous expression of some defense-related genes has been sometimes associated with constitutive expression of defense-related genes, which can have adverse effects on plant growth and development, and usually manifest as shoot and root growth inhibition, necrosis, or both [33, 34]. Such effects directly translate to yield losses. In this study, transgenic bananas expressing *AtEFR* were phenotypically similar to non-transgenic control plants, indicating no limitation to their normal growth characteristics. This further implies that constitutive expression of *AtEFR* in banana is not likely to affect yield, which is consistent with results previously reported in tomato [15, 35], rice [17, 36], potato [18], Medicago [37], apple [19] and sweet orange [20].

Pattern recognition receptors, such as EFR, confer basal resistance against adapted and non-adapted pathogens, and thus contribute to both basal and non-host disease resistances.

The resistance usually is quantitative, but in some instances, it can lead to qualitative or complete resistance [38]. In this study, all the transgenic events showed quantitative resistance manifested by delay in symptom appearance and disease progression, as well as lower disease severity compared with the wildtype controls.

Substantial evidence shows that PRRs confer more broad-spectrum resistance to pathogens compared to R-genes, mainly because they recognize PAMPs, which are often conserved across a wide range of genera and play essential roles in the survival of the pathogens. PRRs are therefore also predicted to confer more durable resistance [32, 39]. However, in rare cases, variations in PAMP genes have been observed in adapted plant pathogens or commensals to evade recognition [40–42]. Also, some pathogens produce effectors that can suppress PTI in some plants [40, 43], but this has not been very effective as some level of PTI is usually retained, as exemplified by enhanced susceptibility to virulent pathogens in mutants of PRRs [44, 45].

EFR sense bacterial EF-Tu through recognition of the N-acetylated elf18 motif, thereby activating PTI [13]. In this study, transgenic bananas expressing *AtEFR* gained responsiveness to elf18 as exhibited by ROS accumulation and upregulation of defense marker genes following elf18 peptide infiltration. These findings indicate that *AtEFR* activity was retained after its transfer to banana and that necessary downstream signaling components are conserved between Arabidopsis and the monocotyledonous plant banana. Accordingly, we observed a significant reduction in the bacterial population at various time points following *Xcm* inoculation compared to control plants, indicating that the activation of early immune outputs (e.g. ROS, defense gene expression) mediated by the recognition of *Xcm* EF-Tu by EFR leads to enhanced resistance to this otherwise adapted pathogen.

Induction of pathogenesis-related (PR) genes is associated with systemic acquired resistance (SAR), a form of plant defense mechanism that begins from a localized region of a plant and spreads throughout the entire plant [46]. Induction of PR gene following elf18 treatment has been reported in transgenic rice expressing *AtEFR* [36]. However, it is not clear whether the upregulation of PR genes observed in this study is a secondary response to constitutive expression of *AtEFR* in banana. Therefore, further studies should be conducted to verify these findings.

In conclusion, this study indicates that *AtEFR* retains its activity in banana when ectopically expressed and confers resistance to *Xcm*. This further confirms that immune signaling networks downstream of PRRs are conserved across various plant families and thus interfamily transfer of PRRs can be used to engineer disease resistance. Given that the resistance observed herein was however quantitative, future studies on banana improvement against bacterial diseases should focus on combining *AtEFR* with other defense genes like *Pflp* and *Hrap*, which could lead to stronger and more durable resistance. The transgenic banana expressing *AtEFR* gene should be further tested under fields conditions for several generations to confirm the durable resistance.

## Supporting information

**S1 Table. List of primers used for qRT-PCR.**
(DOCX)

**S1 Raw images.**
(PDF)

## Author Contributions

**Conceptualization:** Leena Tripathi.

**Data curation:** Jaindra Nath Tripathi, Leena Tripathi.

**Formal analysis:** Mark Adero, Jaindra Nath Tripathi, Leena Tripathi.

**Funding acquisition:** Leena Tripathi.

**Investigation:** Mark Adero, Jaindra Nath Tripathi.

**Methodology:** Jaindra Nath Tripathi.

**Project administration:** Leena Tripathi.

**Resources:** Leena Tripathi.

**Software:** Jaindra Nath Tripathi.

**Supervision:** Jaindra Nath Tripathi, Richard Oduor, Leena Tripathi.

**Validation:** Jaindra Nath Tripathi, Leena Tripathi.

**Visualization:** Jaindra Nath Tripathi.

**Writing – original draft:** Mark Adero, Jaindra Nath Tripathi.

**Writing – review & editing:** Richard Oduor, Cyril Zipfel, Leena Tripathi.

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
