## [Decision Letter · Decision Letter 0]

10 Jul 2023

PONE-D-23-18202Transgenic expression of Arabidopsis ELONGATION FACTOR-TU RECEPTOR (AtEFR) gene in banana enhances resistance against Xanthomonas campestris pv. musacearumPLOS ONE

Dear Dr. Tripathi,

Thank you for submitting your manuscript to PLOS ONE. After careful consideration, we feel that it has merit but does not fully meet PLOS ONE’s publication criteria as it currently stands. Therefore, we invite you to submit a revised version of the manuscript that addresses the points raised during the review process.

We look forward to receiving your revised manuscript.

Kind regards,

Eugenio Llorens

Academic Editor

PLOS ONE

Journal Requirements:

   "This study was supported by the 2 Blades Foundation, the Gatsby Charitable Foundation, and the United States Agency for International Development (USAID)."

   "This study was supported by the 2 Blades Foundation, the Gatsby Charitable Foundation, and the United States Agency for International Development (USAID). "

5. Please include a copy of Table 2 which you refer to in your text on page 24.

Reviewers' comments:

Reviewer's Responses to Questions

**Comments to the Author**

1. Is the manuscript technically sound, and do the data support the conclusions?

Reviewer #1: Yes

Reviewer #2: Yes

Reviewer #3: Partly

2. Has the statistical analysis been performed appropriately and rigorously? 

Reviewer #1: Yes

Reviewer #2: Yes

Reviewer #3: Yes

3. Have the authors made all data underlying the findings in their manuscript fully available?

Reviewer #1: Yes

Reviewer #2: Yes

Reviewer #3: Yes

4. Is the manuscript presented in an intelligible fashion and written in standard English?

Reviewer #1: Yes

Reviewer #2: Yes

Reviewer #3: Yes

5. Review Comments to the Author

Reviewer #1: Line 43: It is suggested to expand on how the disease is spread in the field and to briefly outline the current forms of control.

Line 44: the background of conventional breeding and sources of resistance seems underdeveloped. It is suggested to review the article by Nakato et al. (2019) 'Sources of resistance in Musa to Xanthomonas campestris pv. musacearum, the causal agent of banana Xanthomonas wilt'. Plant Pathology, 68, 49-59. Doi: 10.1111/ppa.12945. These authors report the evaluation of 72 banana accessions representative of Musa diversity for response to BXW by artificial inoculation.

Some of the accessions reported in Nakato et al. (2019) e.g., M. acuminata subsp. zebrina could even be a good reference to contrast the effect of the atEFR gene with existing banana germplasm reported to be resistant/tolerant to BXW.

Line 159: It is suggested to indicate the sublineage of the Xcm colony used in the assay.

Line 179: Please check the expression. In planta bacterial population analysis?

Line 276: The southern indicates the number of insertions in the genome, and although it is correct that it could indicate the number of copies in the genome, it is very risky in the case of a single copy. It would be more correct to say that the number of copies of the transgene incorporated in the different events ranged from at least one to multiple (Fig. 1F). This is simply an observation and is at the discretion of the authors.

Line 337: Please specify the time (days) where complete wilting was observed in all accessions or instead substantiate why a longer time scale was not used for this species which is perennial.

Line 360: Improve the expression of the abscissa legend in Figure 4 C.

Line 390: Please, check the expression. In planta bacterial population analysis was performed...

Line 427: The EFR gene of A. thaliana (AtEFR) was previously overexpressed in several annual and some perennial species. From the point of view of gene efficacy, the assay shown in Table 1 seems more suitable for annual species, not so much in perennials. A more prolonged evaluation in time would seem more appropriate to know the effect of the AtEFR gene in banana cultivation. Moreover, the authors' opinion about their expectations of the durability of resistance in banana cultivation would be desirable.

Line 435: Boschi et al. (2017) is quoted in two different ways: 17 and 36, please leave one alone. Check other quotes, e,g, 19 and 38.

Reviewer #2: Banana Xanthomonas wilt (BXW) is a big threat to banana industry in East and Central Africa. This paper described a reserch on generating transgenic banana plants expressing the AtEFR gene. The authors also evaluated their resistance against XCW under greenhouse conditions. Although the function of EFR gene has been investigated, the application in banana is not carried out yet. Therefore, this research is still meaningful and significant to the banana community. After reviewing, I suggest this paper can be accepted for publication in PLOS ONE. There are some suggestions or comments as below:

1. The scientific name of banana cultivar dwarf Cavendish is needed to be added, Musa spp. Cavendish group AAA.

2. For experimental design, how many plantets per replicate?

3. In Fig 3, the The quality of the figures is not so good for publication. Please try to improve them.

Reviewer #3: The manuscript reports the generation of transgenic lines of banana expressing the EFR receptor from Arabidopsis thaliana and evaluates their resistance against the bacterial pathogen Xanthomonas campestris pv. musacearum (Xcm). The research problem is relevant since banana bacterial wilt caused by Xcm is considered the most devastating disease of banana in East and Central Africa. Conventional breeding for resistance in this crop is hampered by the lack of resistance sources, highlighting the potential of transgenic approaches. The interfamily transfer of the EF-Tu receptor (EFR) from Arabidopsis thaliana has shown to confer PAMP perception and to increase pathogens resistance in several crops in previous studies. The work performed involved the generation and molecular verification of the transgenic lines, the resistance evaluation under greenhouse conditions and indirect verification of the activity of the At EFR receptor through expression analysis of defense-related genes and ROS production after pathogen infection.

Overall, the work is acceptable for publication but structuring of the manuscript should be improved to enhance understanding of the strategy and scope of the activities carried out.

- Include background information on the use of other transgenic approaches in banana for the control of Xcm.

- Why Figure legends are widespread along the manuscript and not necessarily in accordance with the text? Is really confused to follow the reading in this way.

- The term “overexpressing AtEFR” is not correct and is repeated several times in the manuscript. This term is used to refer to gene expression in quantitative and relative terms from one condition to another. In this case the AtEFR gene is not overexpressed in the transgenic lines compared to the control plants (in fact in wt plants the AtEFR gene is not present and therefore not expressed).

- The number and identity of transgenic lines used in the different activities performed varied without any justification or clarification of the selection criteria. Some activities were carried out with all the lines obtained and others with different subsets of selected lines that do not coincide with each other.

- Additional information should be included in resistance assays descriptions: controls used (non-transgenic plants, non-inoculated plants), experimental design (randomized complete design, block design, etc).

- Why AtEFR expression was assessed through both RT-PCR and RT-qPCR?

- Additional information for RT-qPCR analysis is required (primers, cycling conditions, etc).

- Results of the southern blot assay are confused and not clearly presented. It is striking that only one line showed a single copy of the transgene (T5). The other lines showed multiple copies which is not a desirable trait. This fact should be made explicit in the text. Further characterization of these lines is also not justified. A more suitable way for copy number determination should be performed to confirm these results.

- The relative expression of AtEFR gene among transgenic events is part of their molecular verification but is presented separately after the resistance and the ROS assay.

- Figures must be organized logically and referenced in an orderly fashion in the text. Some figures cover more than one essay and are referred to in different sections of the manuscript, which makes it difficult to follow.

- In the case of Figure 1, I suggest separating the E/F panels referring to molecular verification of the lines.

- Figure 3 should also be split in two. One part of the figure refers to the results of the resistance assays, showing the symptoms observed for one of the transgenic lines categorized as PR (T7) compared to the control. The other part of the figure shows the results of the ROS assay performed for the transgenic line T5. It is not understandable why both are shown in the same figure when they refer to different assays and transgenic lines.

- Plant growth was analyzed for different transgenic lines but not for T5, the only one that carries the AtEFR gene in a single copy. This should be performed to complete the characterization of this line.

- Several references are repeated (Lu et al., Lacombe et al., Boschi et al., Piazza et al, Mitre et al., etc). Revise all.

Other minor corrections:

Lane 48: “BB genome”, “B genome”, explain or modify.

Lane 163: “bacterial culture”. You mean “bacterial inoculum” or “bacterial suspension”?

Lanes 163-164: remove “After inoculation” (is repeated at the end of the sentence)

Lane 183: Specify if the leaf-mid rib sections (1 cm) included the inoculation point or not.

Lane 213: “section 4.6”??

Lanes 238-239: “between various transgenic events”…and control non-transgenic plants?

6. PLOS authors have the option to publish the peer review history of their article (what does this mean?). If published, this will include your full peer review and any attached files.

Reviewer #1: No

Reviewer #2: **Yes: **Ou Sheng

Reviewer #3: No

---

## [Author Response · Author response to Decision Letter 0]

5 Aug 2023

Dear Editor and Reviewers, 

Thank you very much for considering our manuscript entitled ‘Transgenic expression of Arabidopsis ELONGATION FACTOR-TU RECEPTOR (AtEFR) gene in banana enhances resistance to Xanthomonas campestris’ for publication in your journal. We are also grateful for your constructive feedback on the manuscript. We have adopted your suggestions and corrected the manuscript with track changes. We now hope that the manuscript meets your expectation. 

Our point-to-point response to the reviewer's comments is as follows:

Reviewer #1

1. It is suggested to expand on how the disease is spread in the field and to briefly outline the current forms of control.

Response: The information has been added in the introduction section. 

2. The background of conventional breeding and sources of resistance seems underdeveloped. It is suggested to review the article by Nakato et al. (2019) 'Sources of resistance in Musa to Xanthomonas campestris pv. musacearum, the causal agent of banana Xanthomonas wilt'. Plant Pathology, 68, 49-59. Doi: 10.1111/ppa.12945. These authors report the evaluation of 72 banana accessions representative of Musa diversity for response to BXW by artificial inoculation. Some of the accessions reported in Nakato et al. (2019) e.g., M. acuminata subsp. zebrina could even be a good reference to contrast the effect of the atEFR gene with existing banana germplasm reported to be resistant/tolerant to BXW.

Response: This information is added to the introduction section. We have also explained how the knowledge from BXW-resistant Musa balbisiana can be transferred to BXW-susceptible banana cultivars.

3. Line 159: It is suggested to indicate the sublineage of the Xcm colony used in the assay.

Response: Xcm isolates from Uganda (sublineage 2) was used in this study. It has been added in the revised manuscript.

4. Line 179: Please check the expression. In planta bacterial population analysis?

Response: It has been revised.

5. Line 276: The southern indicates the number of insertions in the genome, and although it is correct that it could indicate the number of copies in the genome, it is very risky in the case of a single copy. It would be more correct to say that the number of copies of the transgene incorporated in the different events ranged from at least one to multiple (Fig. 1F). This is simply an observation and is at the discretion of the authors

Response: The sentence has been revised accordingly.

6. Line 360: Improve the expression of the abscissa legend in Figure 4 C.

Response: We have revised the figure legends (Figure 4C is now 6C). 

7. Line 390: Please, check the expression. In planta, bacterial population analysis was performed.

Response: It has been revised. 

8. Line 427: The EFR gene of A. thaliana (AtEFR) was previously overexpressed in several annual and some perennial species. From the point of view of gene efficacy, the assay shown in Table 1 seems more suitable for annual species, not so much in perennials. A more prolonged evaluation in time would seem more appropriate to know the effect of the AtEFR gene in banana cultivation. Moreover, the author’s opinion about their expectations of the durability of resistance in banana cultivation would be desirable.

Response: We have added a sentence in the discussion section to state that these transgenic banana will be further evaluated under field conditions for several generations to check the durability of disease resistance. 

9. Line 435: Boschi et al. (2017) is quoted in two different ways: 17 and 36, please leave one alone. Check other quotes, e,g, 19 and 38.

Response: It has been corrected. 

Reviewer #2

1. The scientific name of banana cultivar dwarf Cavendish is needed to be added, Musa spp. Cavendish group AAA.

Response: It has been added. 

2. For experimental design, how many plantets per replicate?

Response: Three replicates were used for each transgenic line, this information has been included in the text. 

3. In Fig 3, the The quality of the figures is not so good for publication. Please try to improve them.

Response: The figures have been revised (Fig 3 in now fig 4) 

Reviwer #3

1. Include background information on the use of other transgenic approaches in banana for the control of Xcm.

Response: It has been added. 

2. Why Figure legends are widespread along the manuscript and not necessarily in accordance with the text? Is really confused to follow the reading in this way.

Response: We have rearranged the figure legends to follow a logical sequence. Please note that the journal requires that we cite the figure legend immediately after its first mention. 

3. The term “overexpressing AtEFR” is not correct and is repeated several times in the manuscript. This term is used to refer to gene expression in quantitative and relative terms from one condition to another. In this case the AtEFR gene is not overexpressed in the transgenic lines compared to the control plants (in fact in wt plants the AtEFR gene is not present and therefore not expressed).

Response: It has been revised.

4. The number and identity of transgenic lines used in the different activities performed varied without any justification or clarification of the selection criteria. Some activities were carried out with all the lines obtained and others with different subsets of selected lines that do not coincide with each other.

Response: The lines generated were subjected to PCR and southern blot analyses. The lines which were PCR positive were acclimatized and evaluated in the greenhouse. Lines for RT-PCR were selected based on their level of BXW resistance in the greenhouse. We selected lines that showed the highest BXW resistance and those that showed the least resistance for qRT-PCR analysis. For bacterial population assay, we selected the line which responded best in the bioassay (T5), the one which showed moderate response (T7), and the non-transgenic control. For the ROS assay and analysis of defense-related genes, we selected the line that had a single gene copy, showed the best response in the glasshouse and qRT-PCR (T5).

5. Additional information should be included in resistance assays descriptions: controls used (non-transgenic plants, non-inoculated plants), experimental design (randomized complete design, block design, etc).

Response: We have added the details. 

6. Why AtEFR expression was assessed through both RT-PCR and RT-qPCR?

Response: RT-PCR was performed to check expression, but further quantitative expression was assessed through qRT-PCR. 

7. Additional information for RT-qPCR analysis is required (primers, cycling conditions, etc).

Response: We have added the additional information. Primer details can be found in Table S1, which have been cited in the relevant sections. 

8. Results of the southern blot assay are confused and not clearly presented. It is striking that only one line showed a single copy of the transgene (T5). The other lines showed multiple copies which is not a desirable trait. This fact should be made explicit in the text. Further characterization of these lines is also not justified. A more suitable way for copy number determination should be performed to confirm these results.

Response: It is indeed true that multiple copy is not a desirable trait. Also, Southern blot is not the most accurate approach for determining copy number. However, it is noteworthy that this is a proof of concept study mainly to determine if AtEFR can function in banana and if it can enhance resistance to EFR. The Southern blot was performed purposely to confirm gene integration and provide some rough information on the copy number. We will further evaluate these lines under field conditions and, if the resistance is sustained, conduct new transformation experiments using prefered banana cultivar in East Africa for product development. At this stage, we will conduct a more elaborate copy number analysis, incorporating other methods, especially because regulators do not prefer lines with multiple copy numbers. 

9. The relative expression of AtEFR gene among transgenic events is part of their molecular verification but is presented separately after the resistance and the ROS assay.

Response: The verification of transgenic events was performed using PCR and Southern and then evaluated for disease resistance. We further checked gene expression in the selective transgenic events with varied resistance levels to check the correlation between expression and disease resistance. Variations in transcript levels were observed between the various transgenic events tested. For example, T5 event showed the highest level of AtEFR gene expression, whereas the lowest gene expression was observed in T3 event.

10. Figures must be organized logically and referenced in an orderly fashion in the text. Some figures cover more than one essay and are referred to in different sections of the manuscript, which makes it difficult to follow. - In the case of Figure 1, I suggest separating the E/F panels referring to molecular verification of the lines. Figure 3 should also be split in two. One part of the figure refers to the results of the resistance assays, showing the symptoms observed for one of the transgenic lines categorized as PR (T7) compared to the control. The other part of the figure shows the results of the ROS assay performed for the transgenic line T5. It is not understandable why both are shown in the same figure when they refer to different assays and transgenic lines.

Response: The figures have been re-arranged to ensure a logical flow.

11. Plant growth was analyzed for different transgenic lines but not for T5, the only one that carries the AtEFR gene in a single copy. This should be performed to complete the characterization of this line.

Response: Plant growth data was taken of eight randomly selected events prior to disease assay to check if expression of AtEFR has any negative impact on plant morphology. We did not observe any significant difference in the growth parameters of the transgenic events tested compared to the control non-transgenic plants. We felt that the eight events aree enough sample size to conclude that expression of AtEFR has no effect on plant growth. Therefore, we did not evaluated additional events for plant growth analysis. Anyways data will be collected for growth and agronomic parameters during confined field trials of the promising events.

12. Several references are repeated (Lu et al., Lacombe et al., Boschi et al., Piazza et al, Mitre et al., etc). Revise all.

Response: It has been corrected.

13. Lane 48: “BB genome”, “B genome”, explain or modify.

Response: BB is genomic group and B is the agenome.

14. Lane 163: “bacterial culture”. You mean “bacterial inoculum” or “bacterial suspension”?

Response: We meant bacterial suspension. This has been revised. 

15. Lanes 163-164: remove “After inoculation” (is repeated at the end of the sentence)

Response: Done

16. Lane 183: Specify if the leaf-mid rib sections (1 cm) included the inoculation point or not.

Response: Yes, the mid-rib sections of the leaves were inoculated with the bacterial suspension.

17. Lane 213: “section 4.6”??

Response: It has been corrected. 

18. Lanes 238-239: “between various transgenic events”…and control non-transgenic plants?

Response: It has been c

---

## [Decision Letter · Decision Letter 1]

18 Aug 2023

Transgenic expression of Arabidopsis ELONGATION FACTOR-TU RECEPTOR (AtEFR) gene in banana enhances resistance against Xanthomonas campestris pv. musacearum

PONE-D-23-18202R1

Dear Dr. Tripathi,

We’re pleased to inform you that your manuscript has been judged scientifically suitable for publication and will be formally accepted for publication once it meets all outstanding technical requirements.

Kind regards,

Eugenio Llorens

Academic Editor

PLOS ONE

Reviewers' comments:

Reviewer's Responses to Questions

**Comments to the Author**

1. If the authors have adequately addressed your comments raised in a previous round of review and you feel that this manuscript is now acceptable for publication, you may indicate that here to bypass the “Comments to the Author” section, enter your conflict of interest statement in the “Confidential to Editor” section, and submit your "Accept" recommendation.

Reviewer #1: All comments have been addressed

Reviewer #2: All comments have been addressed

Reviewer #3: All comments have been addressed

2. Is the manuscript technically sound, and do the data support the conclusions?

Reviewer #1: Yes

Reviewer #2: Yes

Reviewer #3: Yes

3. Has the statistical analysis been performed appropriately and rigorously? 

Reviewer #1: Yes

Reviewer #2: Yes

Reviewer #3: Yes

4. Have the authors made all data underlying the findings in their manuscript fully available?

Reviewer #1: Yes

Reviewer #2: Yes

Reviewer #3: Yes

5. Is the manuscript presented in an intelligible fashion and written in standard English?

Reviewer #1: Yes

Reviewer #2: Yes

Reviewer #3: Yes

6. Review Comments to the Author

Reviewer #1: Each of the comments have been answered satisfactorily and I believe that they have been interpreted correctly.

They have also responded correctly to the comments of other reviewers. This manuscript is considered suitable for publication in the journal.

Reviewer #2: The authors revised according to my previous suggestions. I suggest this paper can be accepted for publication.

Reviewer #3: Authors have adequately addressed the main comments raised in my previous review.

On my point of view the manuscript is now acceptable for publication.

7. PLOS authors have the option to publish the peer review history of their article (what does this mean?). If published, this will include your full peer review and any attached files.

Reviewer #1: No

Reviewer #2: **Yes: **Ou Sheng

Reviewer #3: No

---

## [Editor Report · Acceptance letter]

25 Aug 2023

PONE-D-23-18202R1 

*Transgenic expression of Arabidopsis ELONGATION FACTOR-TU RECEPTOR (AtEFR) gene in banana enhances resistance against Xanthomonas campestris pv. musacearum*

Dear Dr. Tripathi:

I'm pleased to inform you that your manuscript has been deemed suitable for publication in PLOS ONE. Congratulations! Your manuscript is now with our production department. 

Kind regards, 

on behalf of

Dr. Eugenio Llorens 

Academic Editor

PLOS ONE